# Activation of Bradykinin B_2_ Receptors in Astrocytes Stimulates the Release of Leukemia Inhibitory Factor for Autocrine and Paracrine Signaling

**DOI:** 10.3390/ijms252313079

**Published:** 2024-12-05

**Authors:** Ying Lu, Yishan Gu, Anthony S. L. Chan, Ying Yung, Yung H. Wong

**Affiliations:** 1Division of Life Science and the Biotechnology Research Institute, Hong Kong University of Science and Technology, Clear Water Bay, Kowloon, Hong Kong, China; ivy5204053@ntu.edu.cn (Y.L.); yguak@connect.ust.hk (Y.G.); anthonyc@ust.hk (A.S.L.C.); bolisa@ust.hk (Y.Y.); 2School of Public Health, Nantong University, Nantong City 226019, China; 3State Key Laboratory of Molecular Neuroscience, and the Molecular Neuroscience Center, Hong Kong University of Science and Technology, Clear Water Bay, Kowloon, Hong Kong, China; 4Hong Kong Center for Neurodegenerative Diseases, InnoHK, Hong Kong, China

**Keywords:** astrocytes, bradykinin, leukemia inhibitory factor, receptor, signaling

## Abstract

Communications between different cell types within a tissue are often critical for the proper functioning of an organ. In the central nervous system, interactions among neurons and glial cells are known to modulate neurotransmission, energy metabolism, extracellular ion homeostasis, and neuroprotection. Here we showed that bradykinin, a proinflammatory neuropeptide, can be detected by astrocytes, resulting in the secretion of cytokines that act on neurons. In astrocytic cell lines and primary astrocytes, bradykinin and several other ligands acting on G_q_-coupled receptors stimulated Ca^2+^ mobilization, which subsequently led to the release of leukemia inhibitory factor (LIF) and interleukin-6 (IL-6). The bradykinin B_2_ receptor antagonist, HOE-140, effectively blocked the ability of bradykinin to mobilize Ca^2+^ and stimulate mitogen-activated protein kinases (MAPKs) in astrocytes. Interestingly, incubation of neuronal cell lines and primary cortical neurons with conditioned media from bradykinin-treated astrocytes resulted in the activation of STAT3, a key component downstream of LIF and IL-6 receptors. LIF was apparently the major active factor in the conditioned media as the STAT3 response was almost completely neutralized by an anti-LIF antiserum. The presence of kininogen and kallikrein transcripts in neuronal cells but not in astrocytic cells indicates that neurons can produce bradykinin. Correspondingly, conditioned media from neuronal cells stimulated MAPKs in astrocytes in a HOE-140-sensitive manner. These studies demonstrate that paracrine signaling between neurons and astrocytes may involve ligands of G_q_-coupled receptors and cytokines such as LIF.

## 1. Introduction

Neuroinflammation, especially when sustained, has emerged as a prominent factor associated with the progression of many neurodegenerative diseases, including Alzheimer’s disease (AD), Parkinson’s disease (PD), and amyotrophic lateral sclerosis [1,2]. As an immune privileged organ, the brain relies on resident cells, namely astrocytes and microglia, for launching inflammatory responses in the central nervous system (CNS). Astrocytes and microglia need to monitor changes in the microenvironment of the CNS in order to detect and eliminate invading pathogens as well as endogenous materials that are harmful to neurons. Both cell types are intricately involved in the homeostatic regulation of the brain during development, adulthood and aging [3,4,5], a function that requires the orchestration of intercellular communication with neurons and between themselves. Signals arising from neurons are known to activate astrocytes and microglia to release chemokines and pro-inflammatory cytokines (e.g., CCL2, CXCL1, CXCL10, GM-CSF, IL-6, and tumor necrosis factor-α) as well as free radicals (e.g., superoxide and nitric oxide), all of which can, in turn, exert profound effects on neurons [6,7,8]. One of the hallmarks of AD is the progressive accumulation of β-amyloid (Aβ) in senile plaques, wherein Aβ deposition is known to activate astrocytes and microglia [9], which triggers the removal of Aβ by phagocytosis [10]. The accumulation of protein aggregates in other neurodegenerative diseases (e.g., α-synuclein in PD) also stimulates the immune competent cells of the CNS [11]. Sustained activation of astrocytes and microglia is now thought to contribute to synaptic loss, diminished neural function, and neurodegeneration. In addition to neurodegenerative diseases, neuroinflammation is associated with the pathophysiology of other neurological disorders, including stroke, CNS trauma, and chronic pain.

Despite our appreciation of the importance of cellular communication in regulating neuroinflammation, intercellular signaling between different cell types in the brain remains inadequately defined. As one of the most abundant cell types in the brain, astrocytes have long been demonstrated to be essential for the normal functioning of neurons in the CNS [12], and for coordinating the activation of microglia [13]. Astrocytes are well-positioned to perform surveillance of neuronal activity because their foot-like protrusions often wrap around the synapses of neurons [14], while a host of neurotransmitter receptors are expressed in astrocytes to support bidirectional neuron–astrocyte communication. Many neurotransmitters and neuropeptides of physiological importance act on G protein-coupled receptors (GPCRs) that are amply expressed in astrocytes [15]. The notion of astrocytic GPCRs acting as detectors of neuronal signals has previously been confirmed with the demonstration that physiological and pathophysiological changes in neuronal activity can stimulate GPCRs on astrocytes, resulting in Ca^2+^-mediated responses and the release of soluble factors [16,17]. Recent studies have suggested that astrocytic GPCR signaling exhibits neural circuit specificity [18] and may support higher brain functions, such as memory [19]. Intriguingly, a key feature of astrocyte GPCR signaling is the predominant reliance on G_q_-mediated Ca^2+^ mobilization. Ca^2+^ signaling in astrocytes has been proposed to modulate synaptic transmission and plasticity via the release of chemical mediators known as gliotransmitters [17]. Although the precise mechanism by which astrocytic Ca^2+^ signals modulate neuronal activity remains controversial [20], the involvement of G_q_-coupled receptors is indisputable. In a variety of cell types, the activation of G_q_ proteins invariably leads to the release of cytokines and chemokines into the extracellular milieu [21,22], a unique property which is apparently not shared by G_s_ or G_i_ proteins [23]. The ability of G_q_-coupled receptors to induce the release of pro-inflammatory signaling molecules such as interleukin-6 (IL-6) and CXCL8 [23] suggests that these receptors may similarly regulate neuron–astrocyte communication via soluble mediators, in addition to the classical gliotransmitters. This notion is especially tantalizing given the emerging role of GPCRs in the regulation of autocrine and paracrine signaling in various biological systems [24].

Given that numerous neurotransmitters and neuropeptides act on G_q_-coupled receptors in the CNS, it is reasonable to speculate that G_q_-mediated pathways are employed in intercellular communication between neurons and astrocytes. Specifically, neurotransmitters or neuropeptides arising from neurons may act on astrocytic GPCRs to stimulate the release of soluble factors, which subsequently mediate autocrine/paracrine responses that ultimately modulate neuronal function and/or neuroinflammation. Because astrocytes are also known to release growth factors and anti-inflammatory cytokines such as glia-derived neurotrophic factor (GDNF) and IL-10 [25,26], activation of astrocytic G_q_-coupled receptors may regulate functions other than those associated with driving inflammation. Many GPCRs are expressed in astrocytes for gene expression; meanwhile, astrocytic cells are capable of generating secretory cytokines for modulating the activities of neuronal cells. Hence, we are interested in determining whether G_q_-coupled receptor activation in astrocytic cells can produce cytokines of other IL-6 family members (e.g., LIF, CT-1, CNTF, OSM), which in turn act on neuronal cells in a paracrine fashion for communication.

## 2. Results

### 2.1. Activation of G_q_-Coupled Receptors Stimulate the Release of LIF

We have previously demonstrated that G_q_ signals in many cell types are associated with the release of cytokines into the extracellular medium [21,22,23]. Amongst the various cytokines examined, G_q_-induced IL-6 production is particularly robust and appears to drive the subsequent release of pro-inflammatory chemokine CXCL8 by an autocrine loop. Hence, we first explored if other members of the IL-6 family, including cardiotrophin 1 (CT-1), ciliary neurotrophic factor (CNTF), LIF, and oncostatin M (OSM), can also be induced by G_q_ proteins. As glioblastoma/neuroblastoma cell lines and primary astrocytic neuronal cultures are subjected to relatively lower transfection efficiencies, we then followed our previous approach of using HEK293 cells, a frequently used cell model for GPCR/G protein studies with high transfection and expression efficiencies. Different wild-type or constitutively active Gα subunits (harboring a Q → L mutation in the GTPase domain) were transiently expressed and the corresponding culture media removed for the detection of IL-6 family members by ELISA. Besides IL-6, constitutively active QL mutants of Gα_q_ members (Gα_q_, Gα_11_, and Gα_16_) potently stimulated the production of LIF but not ciliary neurotrophic factor, cardiotrophin 1, or oncostatin M (Appendix A). Induction of LIF by constitutively active Gα_q_ mutants was much higher (5–8 fold) than for IL-6. Except for Gα_12_, which weakly but significantly stimulated the release of LIF, constitutively active Gα subunits of the G_s_ (Gα_s_) and G_i_ (Gα_i3_, Gα_oA_, and Gα_z_) subfamilies did not induce the production of the IL-6 family of cytokines. The unique ability of Gα_q_ members to stimulate the release of LIF and IL-6 is further demonstrated by the way in which HEK293 cells transiently express the melatonin MT_1_ receptor with or without Gα_16_ (Appendix A). The G_i_-coupled MT_1_ receptor can activate Gα_16_ but is unable to utilize endogenous Gα_q_ subunits for signal transduction [27]; melatonin is thus expected to stimulate LIF and IL-6 production only in cells co-expressing MT_1_ and Gα_16_. Indeed, application of melatonin at 1 µM for 24 h induced the release of both LIF and IL-6 in MT_1_/Gα_16_ transfectants but not in cells expressing MT_1_ alone (Appendix A). These results reinforce the notion that GPCR-induced cytokine production is specifically mediated by G_q_-dependent pathways.

To determine whether activation of G_q_-coupled receptors in astrocytes can similarly stimulate the release of LIF and IL-6, we screened for the presence of functional G_q_-linked receptors in two glioblastoma/astrocytoma cell lines, U87-MG and 1321-N1. A panel of 23 different GPCR ligands was examined for their ability to stimulate intracellular Ca^2+^ mobilization in a FLIPR platform. The GPCRs were selected according to their known or purported expression in astrocytes [28]. In U87-MG and 1321-N1 cell lines, treatment with saturating concentrations of bombesin (1 μM), bradykinin (1 μM), histamine (10 μM), and sphingosine-1-phosphate (S1P, (1 μM) significantly stimulated intracellular Ca^2+^ responses (Figure 1A). Agonists for muscarinic acetylcholine receptors and endothelin receptors (carbachol (10 μM) and endothelin (100 nM), respectively) also elicited Ca^2+^ responses in 1321-N1 cells, while those for angiotensin II (1 μM), neuromedin U25 (1 μM), oxytocin (1 μM), and purinergic receptors by ATP (1 μM) were effective in U87-MG cells (Figure 1A). Robust Ca^2+^ responses (over 100-fold stimulation) were observed for bradykinin, histamine and carbachol, with the former two occurring in both cell lines, while leukotriene (1 μM), neurotensin (1 μM), orexin (1 μM), and serotonin (10 μM) could not trigger the activity in both cell lines. The lack of a carbachol-stimulated Ca^2+^ response implied the absence of G_q_-coupled muscarinic receptors in U87-MG cells and, hence, carbachol is not expected to stimulate the release of LIF and IL-6 in these cells. Accordingly, treatment of U87-MG cells with 10 µM of carbachol for 24 h failed to induce the production of LIF, whereas incubations with bradykinin (1 μM) or histamine (10 μM) significantly elevated the levels of LIF in the corresponding culture media (Figure 1B). The ability of bradykinin and histamine to induce LIF production was in line with their Ca^2+^ responses in the same cells (Figure 1B). Moreover, the bradykinin-induced LIF production occurred in a dose-dependent manner and maximal stimulation was attained with 24 h of treatment (Figure 1C).

We then prepared primary astrocyte cultures from E16.5–17.5 wild-type C57BL/6J mice to test if LIF production can be similarly regulated by G_q_-coupled receptors in astrocytes. The primary astrocyte cultures were estimated to be >95% pure as revealed by the abundant expression of glial fibrillary acidic protein in the absence of neuronal markers, including NeuN and βIII-tubulin (Appendix A). When a similar panel of ligands was screened against primary astrocytes, ATP (1 μM), bombesin (1 μM), bradykinin (1 μM), carbachol (10 μM), endothelin (100 nM), glutamate (1 μM), neurokinin B (1 μM), oxytocin (1 μM), prostaglandin E_2_ (PGE_2,_ 1 μM), phenylephrine (10 μM), sphingosine-1-phosphate (S1P, 1 μM), and TRAP (1 μM, a protease activating receptor-1 agonist) significantly stimulated Ca^2+^ mobilization (Figure 2A). Bradykinin remained as one of the more effective ligands for Ca^2+^ mobilization in primary astrocytes, although its efficacy and magnitude of stimulation were weaker/lower than that observed in U87-MG cells (Figure 2B); the EC_50_ values for inducing Ca^2+^ signaling by bradykinin in U87-MG cells and primary astrocytes were estimated to be 2 nM and 30 nM, respectively. Treatment of primary astrocytes with bradykinin stimulated the release of LIF in a dose-dependent manner and, as observed in the U87-MG cells, maximal stimulation was attained after around 24 h of ligand treatment (Figure 2C). The EC_50_ for bradykinin-induced LIF secretion (23 nM; Figure 3C) was found to be similar to that observed for Ca^2+^ mobilization (Figure 2B).

### 2.2. Bradykinin-Induced Ca^2+^ Responses in Astrocytic Cells Are Mediated via the B_2_ Receptor

The strong stimulation of LIF secretion by bradykinin in both primary astrocytes and glioblastoma/astrocytoma cell lines prompted us to examine the involvement of specific bradykinin receptor subtypes. Two bradykinin receptors, B_1_R and B_2_R, are known to be expressed in the CNS [29]. The expressions of B_1_R and B_2_R receptors were demonstrated by the Western blotting of cell lysates from U87-MG cells and primary astrocytes (Figure 3A). The level of B_2_R in cultured primary astrocytes was lower as compared with U87-MG cells, which might account for the weaker Ca^2+^ responses to bradykinin seen with the primary cells. Because B_2_R is activated by bradykinin while B_1_R is predominately stimulated by Des-Arg-9-bradykinin, a metabolic by-product of bradykinin, we investigated the involvement of B_2_R using a B_2_R-specific antagonist, HOE-140. As shown in Figure 3B, pre-treatment of U87-MG cells with 5 μM HOE-140 completely abolished the Ca^2+^ signals induced by bradykinin (up to 1 μM). Pre-treatment of U87-MG cells with increasing concentrations of HOE-140 dose-dependently inhibited the bradykinin-stimulated Ca^2+^ response (Figure 3C, upper panel). In primary astrocytes, 0.1 μM and 1 μM HOE-140 had no effect on bradykinin-induced Ca^2+^ transient but the response was suppressed in the presence of 5 μM HOE-140 (Figure 3C, lower panel). However, HOE alone did not affect the Ca^2+^ response in either cell types. These results provide evidence that bradykinin-induced Ca^2+^ signaling was primarily mediated by B_2_R in U87-MG cells and primary astrocytes.

Bradykinin is known to stimulate MAPK pathways in various cell types, including dermal fibroblasts and neural progenitor cells [30,31]. Hence, we assessed the ability of bradykinin to regulate MAPKs in U87-MG cells. The time- and concentration-dependent effects of bradykinin on the stimulatory phosphorylation of extracellular signal-regulated kinase (ERK), p38 MAPK and c-Jun N-terminal kinase (JNK) were examined by Western blot analyses. All three types of MAPKs were activated by 1 μM bradykinin in a time-dependent manner, with the peaks typically occurring around 5–10 min (Appendix A). Bradykinin stimulated the phosphorylation of all three MAPKs at concentrations of 0.1 nM or higher (Figure 3D). The EC_50_ values for inducing phospho-ERK, -p38 and -JNK by bradykinin were estimated to be 0.03 nM, 0.3 nM and 0.3 nM, respectively. Activation of all three MAPKs by 1 µM bradykinin was completely abolished in the presence of 5 μM HOE-140, implicating the involvement of B_2_R (Figure 3E and Appendix A). HOE-140 also blocked the ability of bradykinin to stimulate LIF production in U87-MG cells (Figure 3F). Using a panel of specific protein kinase inhibitors, B_2_R-mediated LIF secretion in U87-MG cells was found to require ERK (PD98059), p38 MAPK (SB203580), and Src kinase (SU6656) (Figure 3F); Src is known to be activated by B_2_R in rat chromaffin medullary cells [32]. Partial inhibition of LIF secretion was observed with the protein kinase C (PKC) inhibitor Go6976 (Figure 3F), indicating the potential involvement of isozymes other than PKCα and PKCβ1. Likewise, the JNK inhibitor SP600125 partially inhibited bradykinin-induced LIF secretion as compared with the control group.

### 2.3. Activation of G_q_-Coupled GPCRs on Astrocytes Leads to the Release of Cytokines That Act as Paracrine Signals for Neurons, as Well as an Autocrine Signal in Astrocytes

Because the cytokines released by astrocytes are likely to act as paracrine signals on neighboring cells, they may activate the corresponding signaling pathways in neurons. To test this hypothesis, U87-MG cells were incubated in the absence or presence of agonists of G_q_-coupled receptors (bombesin, bradykinin, or histamine) for 24 h without serum, the conditioned media were then collected and applied to SK-N-SH neuroblastomas and SK-N-MC neuroepitheliomas for 15 min, and stimulatory phosphorylations of ERK as well as signal transducer and activator of transcription 3 (STAT3) in the recipient cells were assessed by Western blotting (Figure 4A). STAT3 Tyr^705^ phosphorylation was used as a surrogate indicator of cytokine receptor activation, while ERK phosphorylation was used to gauge growth factor signaling. Of the three GPCR ligands tested, bradykinin and histamine treatments consistently produced conditioned media that stimulated STAT3 Tyr^705^ phosphorylation in both SK-N-MC cells and SK-N-SH cells, with little or no change in ERK phosphorylation (Figure 4B). However, conditioned media from bombesin-treated U87-MG cells sometimes failed to induce STAT3 phosphorylation (Figure 4B vs. Figure 4C). The activation of STAT3 in the target cells suggests the secretion of a soluble factor upon G_q_-receptor activation and warrants the validation of LIF being the paracrine factor. The nature of the soluble factor was determined to be heat-sensitive because the STAT3 responses were abolished upon boiling of the conditioned media from agonist-treated U87-MG cells (Figure 4B). The identity of the paracrine factor was further examined by neutralizing the conditioned media with specific antibodies. The incubation of the conditioned media from GPCR ligand-treated U87-MG cells with an anti-LIF antiserum effectively eliminated the STAT3 responses in SK-N-MC cells (Figure 4C). The negative control, anti-GRO antiserum, did not affect the ability of the conditioned media to stimulate STAT3 phosphorylation (Figure 4C). Given that Gα_q_ activation also induced the release of IL-6 (Appendix A), the residual STAT3 activity of the conditioned media following neutralization by anti-LIF antiserum might be attributed to IL-6. Indeed, co-treatment of the conditioned media with anti-LIF and anti-IL-6 antisera completely abolished the STAT3 responses (Figure 4D).

The direct application of LIF to SK-N-MC cells resulted in STAT3 phosphorylation, which occurred in a dose-dependent manner (Figure 4E). LIF also dose-dependently stimulated STAT3 phosphorylation in primary cortical neurons, and the LIF response was apparently stronger than that induced by IL-6 (Figure 4F). The mRNA transcript and protein expression of LIF receptor (LIFR) was detectable in both SK-N-MC cells and primary cortical neurons (Appendix A), suggesting that astrocytic U87-MG cells and primary astrocyte-derived LIF could readily act on the LIFR expressed in SK-N-MC cells and primary cortical neurons for paracrine communication. Similarly, when primary mouse astrocytes were incubated with bradykinin of increasing duration (0, 6, 12, 24 h without serum), the resulting conditioned media were collected and then applied to primary mouse cortical neurons for 15 min. Phosphorylation of STAT3 in the recipient neuronal cells was readily detected (Figure 4G), with the conditioned medium collected from primary astrocytes (having 24 h bradykinin treatment) inducing the highest STAT3 phosphorylation level, while thermo-treatment by boiling significantly diminished the STAT3-activating function of this conditioned medium (Figure 4H). These results demonstrate the presence of an LIF-mediated “Astrocyte → Neuron” paracrine communication. Besides this remote paracrine signaling, GPCR-induced cytokine release may also serve as an autocrine signal within the recipient cells [24]. Because LIF receptor (LIFR) mRNA transcript and protein were also detected in both U87-MG and primary astrocytes (Appendix A), LIF-induced STAT3 phosphorylation was also examined in these cells. Expectedly, we found that both primary astrocytes (Figure 4I) and U87-MG cells could be readily stimulated by LIF in a dose-dependent manner, while LIF was even more potent than IL-6 in stimulating STAT3 phosphorylation in primary astrocytes (Figure 4I). Hence, astrocyte-derived LIF would not only act on neuronal cells in a paracrine manner, but also act on astrocytic cells themselves in an “Astrocyte → Astrocyte” autocrine fashion, as both astrocytes (LIF donors) and neurons (LIF recipients) express functional LIF receptors (Appendix A) for STAT3 signaling. Indeed, such kinds of autocrine–paracrine communication between doner cells and recipient cells have been previously demonstrated for IL-6 [21] and IL-8 [23] between different cell types.

### 2.4. Conditioned Media from SK-N-MC Cells Stimulated MAPKs in U87-MG Cells via B_2_R

Though the presence of kinin-like peptides in the CNS is indisputable, it remains unclear as to whether kinins are solely produced by glia cells. Hence, we assessed whether U87-MG glioblastoma cells and SK-N-MC neuroepithelioma cells express the essential components of the kinin-generating system using RT-PCR. Tissue kallikrein (KLK-1), a serine protease, initiates the cleavage of bradykinin-related peptides from the precursor kininogen (KNG-1). Using human embryonic kidney 293 (HEK293) cells as the positive control, clear discernable bands corresponding to KNG-1 and KLK-1 transcripts could be identified (Figure 5A). KNG-1 and KLK-1 transcripts were likewise detected in SK-N-MC cells (Figure 5A). However, only faint bands corresponding to KNG-1 and KLK-1 transcripts were observed in U87-MG cells, and the levels of these were much lower as compared with those of SK-N-MC cells (Figure 5A). We found that KNG-1 and KLK-1 transcripts are present at very low levels in U87-MG but are more abundant in SK-N-MC cells. Because bradykinin-induced MAPK activation via B_2_R has been demonstrated in astrocytes (Figure 3D–E), to establish if neurons can exert paracrine regulation on astrocytes via bradykinin, we further examined the effect of conditioned medium from SK-N-MC (i.e., CM-SK) on ERK, p38, and JNK activation in U87-MG. Conditioned media from cells freshly cultured for different periods (15 min, 1 h, 3 h and 6 h) were applied to serum-starved U87-MG cells for 10 min in the absence or presence of HOE-140, and U87-MG cell lysates were then subjected to Western blot analyses. As compared with the control U87-MG cells, incubated with normal culture medium for 10 min, the levels of phosphorylated ERK, p38, and JNK were found to be significantly elevated upon treatment of U87-MG cells with CM-SK at all three examined time points (Figure 5B–D). Stimulatory phosphorylations of MAPKs by CM-SK were significantly inhibited in the presence of HOE-140 (Figure 5B–D). These results indicate that the induction of MAPK activation in U87-MG cells by CM-SK were very likely to be mediated via bradykinin released from SK-N-MC cells. However, it should be noted that sources of biological GPCR agonists other than bradykinin (e.g., histamine as shown in Figure 1B), or bradykinin derived from other tissue cells, may also be capable of initiating LIF production from astrocytes.

## 3. Discussion

Gliotransmission, an active communication between astrocytes and neurons, has been proposed to play a critical role in modulating the functions of synapses, neurons, and neural circuits [17]. It is generally accepted that astrocytes receive neuronal information via a wide array of mechanisms and translate it into a complex intracellular Ca^2+^ signal for processing. Among the various GPCRs that are endogenously expressed in astrocyte [33,34], many are classified as G_q_-coupled receptors for neurotransmitters and neuropeptides that regulate Ca^2+^ signaling. Here we have illustrated for the first time that, within the G protein family, members in the G_q_ group seem to be the most effective in terms of the robust induction of LIF (as well as IL-6, though to a lesser extent). These G_q_-mediated activities are relatively specific, as only LIF and IL-6, but no other LIF-related cytokines (i.e., CT-1, OSM, and CNTF), were produced upon G_q_ activation. We have further demonstrated that the functional coupling between G protein-coupled melatonin receptor and G_16_ (a G_q_ member) in transfected HEK293 cells are also capable of efficiently inducing LIF production. Coincidently, our early work on G protein-mediated transcription factor activations has also suggested the major role of G_q_ family members in inducing STAT3 phosphorylation [35], a stimulatory event that is shared with the signaling pathway of LIF. When this G_q_-mediated LIF production event was examined in human astrocytic cell lines (U87-MG and 1321-N1) and mouse primary astrocytes, treatments with bradykinin and other biological ligands produced typical G_q_-mediated intracellular Ca^2+^ transients, and the subsequent dose- and time-dependent LIF production could be observed. It should be noted that 1321N1 astrocytoma cells did not respond to ATP for triggering Ca^2+^ transients (a well-characterized signal for astrocytes), this finding illustrates the limitation of this model cell line, and demonstrates the necessity of primary cultures for parallel investigation, as performed in the current report. The bradykinin-triggered Ca^2+^ transient and MAPK (ERK, JNK, and p38) activations were all blocked by HOE-140 (a specific antagonist for bradykinin type II receptor, B_2_R). Astrocyte-derived LIF production upon bradykinin treatment was also consistently suppressed by the targeted inhibitions of B_2_R, Src kinases, and various MAPK subgroups. G_q_-mediated MAPK activations have been well characterized, and our group has previously clarified that several G_q_-coupled receptors (including B_2_R) rely on initiating the Src and Ca^2+^ activities which subsequently trigger various subgroups of MAPKs to activate transcription factors [36,37]. Hence, it is not surprising that bradykinin is able to utilize the same signaling pathway for the transcriptional induction of LIF. In fact, many cytokines (e.g., IL-6, GM-CSF) and chemokines (e.g., IL-8) have had their inductions demonstrated, inductions which are effectively triggered by expressing activated mutants of the G_q_ family members, or acting through endogenous receptors coupled with G_q_ family members directly [21,23].

The intricate association between astrocytes and neurons is crucial for the maintenance of brain homeostasis [38]. In our current study, the ability of bradykinin to induce astrocyte-derived LIF raises the question of whether the induced LIF molecules would act on astrocytic cells themselves as an “autocrine” pathway or on a different neighboring cell type, for example, neuronal cells, in a “paracrine” manner. As the overnight treatment of bradykinin on astrocytic U87-MG cells was found to result in the production of secretory LIF, we hence examined whether the resulting LIF-containing conditioned medium could act on neuronal cells so as to trigger any LIF-mediated intracellular signals, such as the Tyr^705^ phosphorylation of STAT3. We successfully showed that the astrocyte-derived conditioned medium could effectively stimulate Tyr^705^ phosphorylation of STAT3 in both SK-N-MC cells and SK-N-SH cells, and that this STAT3 phosphorylation level could be diminished by boiling or with the use of LIF-neutralizing antibodies, indicating that the major inducing factor in the astrocyte-derived conditioned medium should be a temperature-sensitive LIF protein. Furthermore, the presence of mRNA transcript and the protein of LIFR in U87-MG cells and primary astrocytes, as well as in SK-N-MC cells and primary neurons, serves as additional evidence that astrocyte-derived LIF protein molecules can act on astrocytic cells themselves (autocrine) and on neuronal cells (paracrine) simultaneously (Figure 6). It should be noted that astrocytic cultures are likely to contain microglia, though whether they play a modulatory role for the bradykinin-mediated astrocyte–neuron interaction remains to be determined. LIF has long been recognized for its regulatory effects on astrocyte development via its stimulation of the maturation of astrocytes from astrocytic progenitor cells [39]. Astrocyte-derived LIF expands the neural stem/progenitor pool following perinatal hypoxia–ischemia [40]. Moreover, the neuroprotective effect of LIF against brain injuries is obtained through the stimulation of the astrocytes that prevent secondary neurogenerative responses [41]. On the other hand, LIF appears to promote the self-renewal of neural stem cells (NSCs) so as to expand the NSC pool [42]. Its stimulatory effect on neural precursor cells (NPCs) is important for the regeneration of neurons after spinal cord and brain injuries [43]. Furthermore, oligodendrocytes and their progenitor cells can be stimulated by either astrocyte-derived or exogenous LIF so as to initiate oligodendrocyte proliferation and remyelination [44,45]. Therefore, astrocyte-derived LIF is likely to elicit widespread effects on various cell types within the nervous system, including, but not limited to, (i) autocrine effects for astrocyte maturation, (ii) paracrine stimulation for neuron regeneration, and (iii) paracrine modulation for oligodendrocytes toward myelination.

Bradykinin-mediated actions are not limited to arterial pressure control and inflammation, as it may also participate in neurotransmission as a neuropeptide, with its cellular signaling in favor of neurogenesis over gliogenesis [46]. Our current study has suggested that bradykinin, by acting through its G_q_-coupled receptors (i.e., B_2_R) that are endogenously expressed in astrocytes, could link to the generation of secretory cytokines, including LIF, which in turn would act on both astrocytic cells and neuronal cells. Indeed, bradykinin is widely distributed in the periphery and brain. Several studies on the neuro-modulatory actions of bradykinin and B_2_R indicate that this neuropeptide plays an important role for neural fate determination [33]. Our results show that the mRNA transcripts of KNG-1 and KLK-1 are readily detected in SK-N-MC neuroepithelioma cells, rather than in U87-MG glioblastoma cells. As KLK-1 is the serine protease responsible for the generation of bradykinin-related peptides from KNG-1, it is reasonable to suggest that neuronal cells could be one of the major sources for bradykinin within the nervous system. Release of bradykinin by neurons may be a constitutive process, even if stimulatory release of this neuropeptide should not be ruled out. Our finds support the idea that conditioned media from SK-N-MC cells (collected 15 min to 6 h after serum removal) could effectively trigger the MAPK (ERK, JNK, and p38) activation of U87-MG cells in an HOE-140 (B_2_R antagonist) sensitive manner. Previous studies have suggested that high concentrations of bradykinin and increased B_2_R activity can be found in the cortex of Alzheimer’s disease patients, indicating the participation of this receptor signaling pathway in inflammation-related processes [47]. However, bradykinin may induce anti-inflammatory and neuroprotective effects in the brain, mediated by glial cells, as the activation of B_2_R in astrocytes has been demonstrated to produce prostaglandin E2, with the consequential increase in intracellular cAMP leading to decrease levels of TNF-α and IL-1β [48,49]. Further evidence is given by the neurogenesis-promoting actions of bradykinin/B_2_R signaling in neural and pluripotent stem cell models [50], in which in situ hybridization revealed the presence of B_2_R mRNA throughout the nervous system in mouse embryos. Here, neurogenesis was augmented by bradykinin in the middle and late stages of the differentiation process, in turn implying that bradykinin/B_2_R signaling acts as a switch for neural fate determination.

SK-N-MC cells (neuroepithelioma), SK-N-SH cells (neuroblastoma), U87-MG cells (glioblastoma) and 1321N1 cells (astrocytoma) are all human cancer cell lines of neuronal/astrocytic origins. They have been extensively utilized for several decades in cell biology investigations and have been involved in the establishment of several functional characterization studies of neuronal and astrocytic cells; however, they can be considered as “models” of neuronal and astrocytic cells only, as they are neither native neurons nor native astrocytes. This is the reason that these cell lines were utilized in our early study: in order to explore the possible role of G protein-coupled receptor (GPCR)-mediated LIF production from U87-MG glioblastoma cells and the subsequent LIF action on SK-N-MC neuroepithelioma cells and SK-N-SH neuroblastoma cells. We then utilized primary neuronal and astrocytic cultures to provide further supportive data for our hypothetic model. The use of human cell lines has typically insisted on utilizing lines of “human origin” for fundamental studies, with the drawback of possible genetic instability associated with the potential cancerous backgrounds of these lines; however, the use of primary neuronal and astrocytic cultures of mice has enabled a higher biological relevance of investigations, though allowing for the encounter of the possible species-specificities of the various biological activities. Therefore, though the use of glioblastoma cells and neuroblastoma cells may accelerate the early progress of a relevant investigation, further experimental proofs with primary neuronal and astrocytic cultures are required.

Taken together, our current study has illustrated that neuronal-cell-derived bradykinin may stimulate astrocytic cells for the robust generation of secretory LIF and that the resulting LIF proteins may act on endogenous LIFR expressed in both astrocytic cells (autocrine) and neuronal cells (paracrine). Such bidirectional interactions illustrate the intimate communication between astrocytes and neurons for bradykinin-induced LIF/STAT3 signaling propagation in the mammalian nervous system (Figure 6). Further investigative approaches may include the more biologically relevant “co-culture” models of astrocytic cells and neuronal cells, as well as the possible paracrine communication of microglia cells with either astrocytic cells or neuronal cells, as reflected by the complexity of signal networking among multiple cell types in the nervous system.

## 4. Materials and Methods

### 4.1. Materials

Human cell lines, including HEK293 (embryonic kidney), U87-MG (glioblastoma), SK-N-SH (neuroblastoma) and SK-N-MC (neuroepithelioma) cells, were purchased from the American Type Culture Collection (Rockville, MD, USA), while 1321N1 (astrocytoma) cells were obtained from Merck (Burlington, MA, USA). Anti-β-actin antibody and HOE-140 were obtained from Sigma-Aldrich (St. Louis, MO, USA). Antisera against the stimulatory phosphorylated form(s) or the total amounts of STAT3, ERK, JNK, and p38 were purchased from Cell Signaling Technology (Danvers, MA, USA). Anti-B_1_R and anti-B_2_R were obtained from Santa Cruz Biotechnology (Santa Cruz, CA, USA). Neutralizing antibodies for LIF, IL-6, and GRO were obtained from R&D Systems, all of which were applied at 5 µg/mL in cell culture medium for neutralizing purposes. Cell culture reagents, including trypsin, fetal bovine serum, penicillin–streptomycin mixture, minimum essential media (MEM), and Dulbecco’s modified Eagle medium (DMEM), as well as the Fluo-4 reagents for Ca^2+^ detection by FLIPR were obtained from Life Technologies (Carlsbad, CA, USA). Trizol reagent and cDNA synthesis kits for RT-PCR were obtained from Invitrogen (Carlsbad, CA, USA). KAPA HiFi HotStart Ready Mix was obtained from Kapa Biosystems (Wilmington, MA, USA). cDNAs encoding different G protein alpha subunits (Gα), including both the wild type (WT) and the constitutively activated (QL) mutant of any Gα subunits, were purchased from the cDNA Resource Center (Bloomsburg, PA, USA). Transfection of these Gα subunit cDNAs to HEK293 cells was performed using Lipofectamine^TM^ 2000 transfection reagents according to the supplier’s instruction (Thermo Fisher Scientific, Waltham, MA, USA). All of the GPCR ligands used in the current research project were obtained from Tocris Bioscience (Minneapolis, MN, USA) and Sigma-Aldrich (St. Louis, MO, USA).

### 4.2. Cell Culture

HEK293, U87-MG, 1321N1, SK-N-SH, and SK-N-MC cells were cultured in MEM supplemented with 10% (vol/vol) fetal bovine serum (FBS), 50 U/mL penicillin, and 50 μg/mL streptomycin. Primary astrocyte cultures were prepared from the cerebral cortexes of neonatal mice (P2) [51]. The cerebral hemispheres were isolated and transferred to ice-cold Hank’s buffer, and the meninges were carefully removed. Tissues were then minced into approximately 1 mm pieces, triturated, filtered through a 60 µm nylon screen, and collected by centrifugation at approximately 3000× *g* for 5 min. The cell pellets were dispersed with a pipette and resuspended in a medium containing 10% (vol/vol) FBS in low-glucose Dulbecco’s modified Eagle’s medium. After trituration, the cells were filtered through a 10 µm screen and then plated into 6-well plates at a density of 3 × 10^5^ cells/mL and cultured for about 10 days. The medium was replaced twice per week. Dibutyryl cyclic adenosine monophosphate (DBcAMP; 0.15 mM, Sigma, St Louis, MO, USA) was added to induce differentiation when the cells were grown to 95% confluence. Primary mouse cortical neuronal cultures were obtained from E16 ICR mouse embryos [52]. Briefly, cerebral cortices were removed from mouse embryos and mechanically dissociated in PBS with glucose (18 mM). Neurons were then seeded onto poly-L-lysine (1 mg/mL)-coated 96-well plates at 5 × 104 cells/well in a B27 Plus Neuronal culture system (Thermo Fisher Scientific), supplemented with 2 mM glutamine, 5 μM mercaptoethanol, 100 U/mL penicillin and 100 μg/mL streptomycin. The cultured neurons were maintained in a humidified incubator at 37 °C with 5% CO_2_ for 7 days prior to use.

### 4.3. Conditioned Media Preparation

U87-MG cells (80–100% confluence) in 6-well plates were treated individually with 1 mL of MEM (without FBS) per well containing bombesin (1 μM), bradykinin (1 μM), or histamine (10 μM) for 24 h, and the resulting conditioned media were collected in 1.5 mL microcentrifuge tubes. These conditioned media were either immediately applied to SK-N-SH/SK-N-MC cells (80–100% confluence) in 6-well plates for 15 min, or subjected to heat treatment on a 96 °C heat block for 5 min, and then cooled to 37 °C before they were applied to the SK-N-SH/SK-N-MC cells for 15 min. Preparation of conditioned media from bradykinin-treated primary astrocytes was performed in the same way and subsequently applied to primary neurons for 15 min. For the SK-N-MC-derived conditioned media, SK-N-MC cells (80–100% confluence) in a 10 cm culture dish were serum-starved in 10 mL of MEM (without FBS), with 1 mL of the resulting conditioned medium being collected at indicated time points (15 min, 1 h, 3 h, 6 h). Simultaneously, U87-MG cells (80–100% confluence) in 12-well plates were serum-starved in MEM for 6 h, followed by pretreatment of HOF-140 (5 μM, 15 min), and then challenged with the SK-N-MC-derived conditioned media (mentioned above) for 10 min before cell lysis. Cell lysates were subjected to SDS-PAGE and Western blot for the detection of stimulatory phosphorylated forms and the total amounts of various MAPK subtypes (ERK, JNK, and p38 MAPK).

### 4.4. Cytokine Detection

Conditioned media were analyzed for the presence of multiple cytokines by using the cytokine multiplex assay kits (Merck Millipore, MA, USA) as previously described [21,23]. Briefly, a bead mixture with various capture antibodies (50 μL) was added to a 96-well microtiter plate, followed by 50 μL of cytokine standard mix or test sample and then 25 μL of a biotinylated detection antibody mixture. The mixtures were incubated for 30 min at 25 °C with gentle shaking. Unbound antibodies were removed by three gentle washes. Finally, 50 μL of streptavidin–PE solution was added and incubated for another 30 min with gentle shaking. The detection beads were then washed again and resuspended in 120 μL of reading buffer. The Bio-Plex™ 200 system (Bio-Rad Laboratories, Hercules, CA, USA) with the Bio-Plex manager software (version 5.0) was utilized for signal detection and data analysis. Curve fitting was applied to each standard curve according to the manufacturer’s manual and sample concentrations were interpolated from the standard curves. Individual detections for LIF and IL-6 were performed using the corresponding ELISA kits from Abcam (Cambridge, UK) according to the manufacturer’s instructions.

### 4.5. Neutralization of Cytokines with Anti-Sera

For the neutralization of specific cytokines, each conditioned medium obtained from U87-MG cells (pretreated 24 h with or without 1 μM bombesin, 1 μM bradykinin, or 10 μM histamine) were divided into 4 equal portions, followed by 15 min incubation with (i) no neutralizing antibody, (ii) 10 μg/mL of Anti-GRO, (iii) 10 μg/mL of Anti-LIF, and (iv) 10 μg/mL of Anti-LIF plus 10 μg/mL of Anti-IL6. The resulting conditioned media were applied to serum-starved SK-N-MC cells (80–100% confluence) in 12-well plates for 15 min, with the subsequent cell lysates subjected to SDS-PAGE and Western blot with specific antibodies in order to detect stimulatory phosphorylated forms and the total amounts of STAT3.

### 4.6. Pretreatment on Experimental Cells with Specific Inhibitors and Antagonists

Experimental cells (80–100% confluence) in 96-well plates (black-walled with a transparent bottom) were pretreated with HOE-140 (5 μM) for 15 min prior to stimulation with bradykinin (1 μM). The resulting Ca^2+^ signal profiles were monitored by FLIPR for 3 min. Additionally, experimental cells (80–100% confluence) that had been serum-starved with MEM without FBS in 12-well plates were pretreated with various inhibitors (of specific targets) 30 min prior to bradykinin stimulation, as follows: 10 μM of Go6976 (protein kinase C); 10 μM of SB203580 (p38 MAPK); 10 μM of SU6656 (Src kinases); 10 μM of SP600125 (JNK); 10 μM of PD98059 (ERK); and 5 μM of HOE-140 (bradykinin B2 receptor). Bradykinin (1 μM) was subsequently added to trigger B2 receptor activation for 10 min and the resulting cell lysates were subjected to SDS-PAGE and Western blot with specific antibodies which recognize the stimulatory phosphorylated forms and the total amounts of various MAPK subtypes (ERK, JNK, and p38 MAPK). Alternatively, cells subjected to pretreatment with the same series of specific inhibitors were stimulated with bradykinin for 24 h, followed by detection of induced LIF production with ELISA assays.

### 4.7. Measurement of Intracellular Ca^2+^ Transients by Fluorometric Imaging Plate Reader (FLIPR)

Cells were seeded in 96-well plates (clear bottom and black wall) at 2.5 × 10^4^ cells per well on the day before assay. The cells (~80–100% confluence) were then washed with Hank’s balanced salt solution (HBSS) and aspirated, followed by addition of 100 μL of FLIPR labeling dye (containing Fluo-4 and probenecid in HBSS) to each well before incubation at 37 °C for 1 h. Finally, the 96-well plate containing the labeled cells was transferred to the FLIPR (Molecular Devices, San Jose, CA, USA), and 50 μL of HBSS (with or without agonists) was added to each well. The fluorescent signals that reflect the intracellular Ca^2+^ transients were monitored by an excitation wavelength of 488 nm, and detection was undertaken with an emission wavelength from 510 to 570 nm for 3 min upon the drug treatments, as described previously [23].

### 4.8. RT-PCR

For mRNA detection, the total RNA of the cells was extracted using Trizol reagent. One microgram of total RNA was reversely transcribed using cDNA synthesis kits for reverse transcription PCR (RT-PCR) according to the manufacturer’s protocol. The cDNA was amplified using the following primers: KNG-1-forward (5′-TGCTCCAGGCTGCTACTAAGT-3′), KNG-1-reverse (5′-GGCTTCAGTTATGCGGTACAA-3′), KLK-1-forward (5′-GGGTCGCCACAACTTGTTTG-3′), KLK-1-reverse (5′-GCTGTAGTCCTCGTCTGCTT-3′), GAPDH-forward (5′-AAGTTGTCATGGATGACCTTGGC-3′), and GAPDH-reverse (5′-GGCGTCTTCACCACCATGGAG-3′). The PCR amplifications were performed at 95 °C for 3 min, followed by 35 cycles of thermal cycling at 98 °C for 20 s, 60 °C for 15 s and 72 °C for 15 s, and a final extension at 72 °C for 15 s. GAPDH was used as an endogenous control to normalize differences. After PCR amplification, 5 µL of the PCR products were used for electrophoresis on 1.5% agarose gels and stained with Midori Green. All primers utilized in our current study were designed using Primer-BLAST (https://www.ncbi.nlm.nih.gov/tools/primer-blast/index.cgi?GROUP_TARGET=on accessed on 2 December 2024) (NIH, Bethesda, MD, USA).

### 4.9. Responsiveness of Experimental Cells Towards LIF and IL-6

To determine if the experimental cells were readily responsive to LIF and IL-6, SK-N-MC cells, mouse primary neurons and primary astrocytes (80–100% confluence in 12-well plates) were serum-starved for 6 h, followed by treatment of indicated concentrations (0–100 ng/mL) of LIF and IL-6 for 15 min. The subsequent cell lysates were subjected to SDS-PAGE and Western blot with specific antibodies to detect stimulatory phosphorylated forms and the total amounts of STAT3.

### 4.10. SDS-PAGE and Western Blot

SDS-PAGE and Western blotting were performed as described in our previous studies [21]. In brief, cells in 6-well plates (5 × 10^5^ cells/well) were lysed in 500 μL of ice-cold lysis buffer (50 mM Tris-HCl at pH 7.5 containing 100 mM NaCl, 5 mM EDTA, 40 mM NaP_2_O_7_, 1% Triton X-100, 1 mM dithiothreitol, 200 μM Na_3_VO_4,_ 100 μM phenylmethylsulfonyl fluoride, 2 μg/mL leupeptin, 4 μg/mL aprotinin, and 0.7 μg/mL pepstatin) and then gently shaken for 30 min. The amount of protein in each sample was determined by detergent compatible (DC) protein assay according to the supplier’s instruction (Bio-Rad, Hercules, CA, USA). Clarified lysates (30 μg protein per sample) were resolved on 12% SDS-polyacrylamide gels and then transferred to nitrocellulose membranes (Westborough, MA, USA). Resolved proteins were detected by specific primary antibodies and horseradish peroxidase-conjugated secondary antisera. The level of β-actin was determined as an internal control. Immunoblots were visualized by chemiluminescence with the WesternBright ECL reagents from Advansta (San Jose, CA, USA). Chemiluminescence signals were then detected by ChemiDoc imaging systems (Bio-Rad, Hercules, CA, USA) with the subsequent band densities quantified by ImageJ (https://imagej.net/ij/) analysis software (National Institutes of Health, MD, USA). All primary antibodies which specifically recognize B_1_R, B_2_R, ERK, p-ERK, p38, p-p38, JNK, p-JNK, STAT3, p-STAT3, and β-actin were applied at 1:1000 dilution for Western blotting.

### 4.11. Statistical Analysis of Experimental Data

Experimental data from all assays were statistically analyzed by GraphPad Prism version 7.0 for PC (GraphPad Software, La Jolla, CA, USA). The data were analyzed using one-way analysis of variance (ANOVA) test, then further compared with basal or control group using Dunnett’s test. *p*-values of less than 0.05 were defined as statistically significant. Independent experiments were typically performed in triplicate with the number of repeats (*n* values) indicated in the corresponding figure legends.

### 4.12. Research Resource Identifiers (RRID)

The ID numbers of key resources (antibodies and cell lines) refer to their research resource identifiers (RRID) and are provided here for the accurate identification of materials, as follows: U87-MG (RRID:CVCL_0022); 1321N1 (RRID:CVCL_0110); SK-N-MC (RRID:CVCL_0530); SK-N-SH (RRID:CVCL_0531); HEK293 (RRID:CVCL_0045); STAT3 antibody (RRID:AB_331588); phospho-STAT3 (Tyr705) antibody (RRID:AB_331586); p44/42 MAPK (Erk1/2) antibody (RRID:AB_330744); phospho-p44/42 MAPK (ERK1/2) (Thr202/Tyr204) antibody (RRID: AB_331646); SAPK/JNK antibody (RRID:AB_2250373); phospho-SAPK/JNK (Thr183/Tyr185) antibody (RRID: AB_331659); P38 MAPK antibody (RRID: AB_330713); phospho-p38 MAPK (Thr180/Tyr182) antibody (RRID: AB_2139682); anti-hIL-6 neutralizing antibody (RRID: AB_354281); anti-hLIF neutralizing antibody (RRID: AB_2135976); anti-hGRO neutralizing antibody (RRID: AB_2292460).

## Figures and Tables

**Figure 1 ijms-25-13079-f001:**
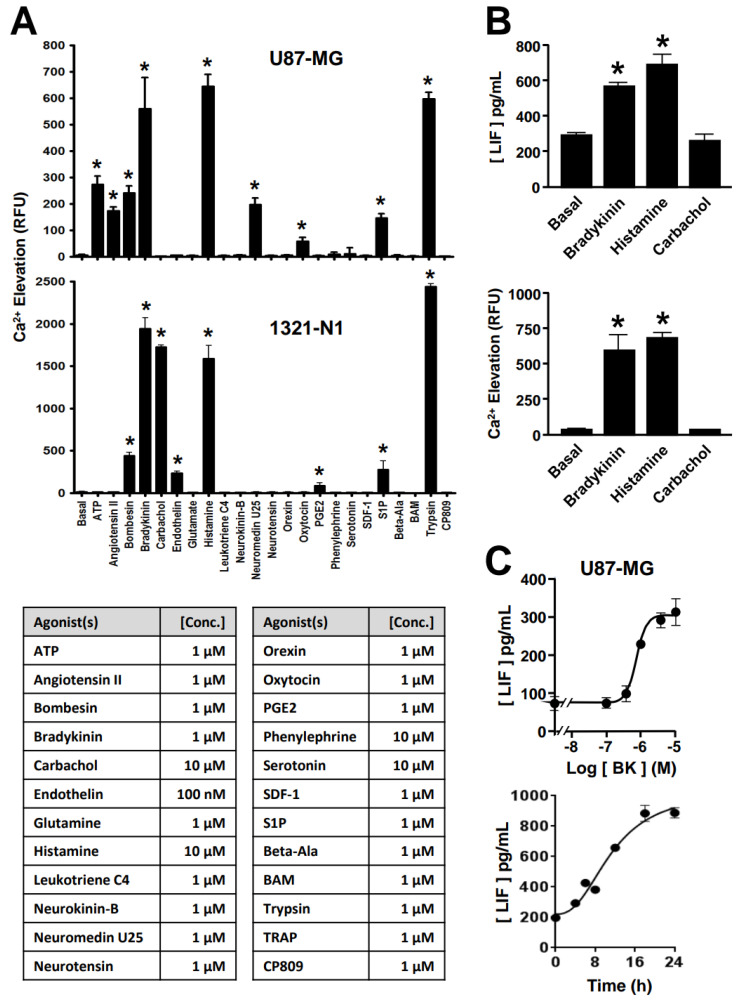
G_q_-coupled receptors stimulate Ca^2+^ mobilization and LIF production in astrocytic cells. (**A**) U87-MG and 1321-N1 cells were challenged with a panel of G_q_-coupled receptor agonists and assayed for intracellular Ca^2+^ elevation. RFU, relative fluorescent unit. (**B**) U87-MG cells were stimulated with the indicated GPCR agonists to trigger Ca^2+^ elevation. These cells were also stimulated by selected agonists, followed by detection of LIF in the conditioned media. (**C**) Bradykinin-induced LIF secretion occurred in a time- and agonist-concentration-dependent manner. U87-MG cells were treated with various concentrations of bradykinin for 24 h and the level of LIF in the conditioned media determined by ELISA (upper panel). Cells were treated with 1 µM of bradykinin and the conditioned media were collected at different time points for LIF ELISA assay (lower panel). * GPCR agonists significantly triggered intracellular Ca^2+^ elevation or LIF production (* *p* < 0.05; *n* = 3).

**Figure 2 ijms-25-13079-f002:**
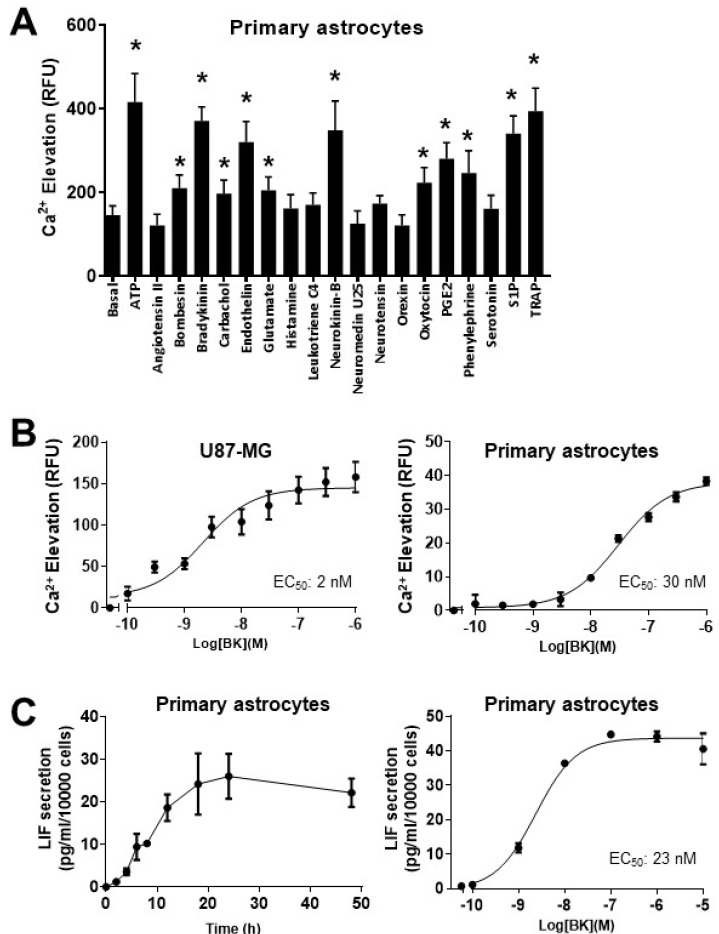
Activation of G_q_-coupled receptors stimulated Ca^2+^ mobilization and LIF production in primary cortical astrocytes. (**A**) Primary astrocytes were challenged with different agonists for G_q_-coupled receptors and assayed for intracellular Ca^2+^ elevation as in Figure 1A. * GPCR agonists significantly triggered intracellular Ca^2+^ elevation (* *p* < 0.05; *n* = 4). (**B**) Bradykinin (BK) stimulated Ca^2+^ mobilization in primary astrocytes in a dose-dependent manner, with an EC_50_ value similar to that of U87-MG cells. (**C**) Bradykinin-induced LIF secretion by primary astrocytes occurred in a time- and agonist-dose-dependent manner. Primary astrocytes were assayed as in Figure 1C.

**Figure 3 ijms-25-13079-f003:**
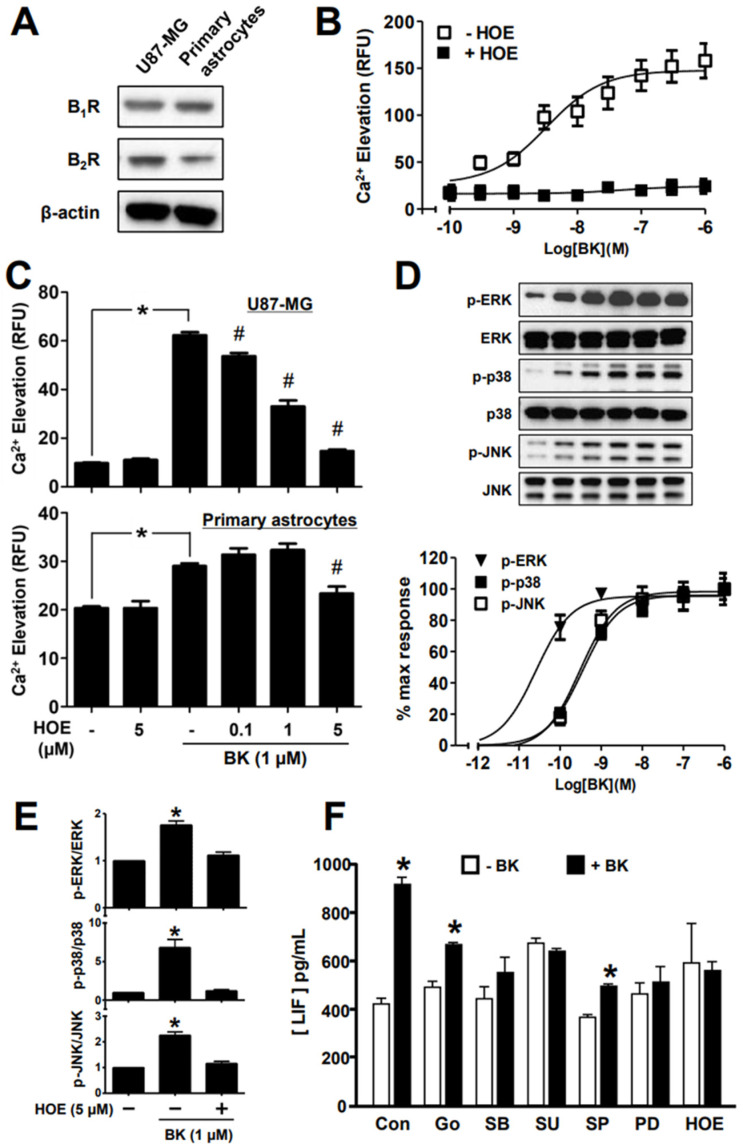
Bradykinin stimulated Ca^2+^ mobilization and activated MAPK through the bradykinin B2 receptor (BK_2_R) in U87-MG cells and primary astrocytes. (**A**) Cell lysates from U87-MG cells and primary astrocytes were immunoblotted with antibodies specific for bradykinin B1 and B2 receptors. (**B**) U87-MG cells were subjected to the FLIPR assay to measure Ca^2+^ transients induced by increasing concentrations of BK (from 0.1 nM to 1 μM) in the presence or absence of the BK_2_R-selective antagonist HOE-140 at 5 μM. (**C**) U87-MG cells and primary astrocytes were assayed for bradykinin-induced Ca^2+^ mobilization in the absence or presence of increasing concentrations of HOE-140. (**D**) U87-MG cells were treated with different concentrations of bradykinin for 10 min and cell lysates were probed with phospho-specific antibodies against ERK, *p*-38 MAPK, or JNK; band intensities were quantified by Image J and are shown in the lower panel. (**E**) U87-MG cells were stimulated with 1 µM bradykinin in the absence or presence of 5 µM HOE-140, and the phosphorylation states of the MAPKs were determined as shown in panel D. (**F**) U87-MG cells were treated with 1 µM bradykinin with or without specific protein kinase inhibitors. Go, Go6976 (protein kinase C); SB, SB203580 (p38 MAPK); SU, SU6656 (Src kinase); SP, SP600125 (JNK); PD, PD98059 (ERK). * Bradykinin significantly triggered Ca^2+^ elevations and stimulated the LIF production (* *p* < 0.05; *n* = 4). # HOE-140 significantly inhibited the bradykinin-induced Ca^2+^ elevations.

**Figure 4 ijms-25-13079-f004:**
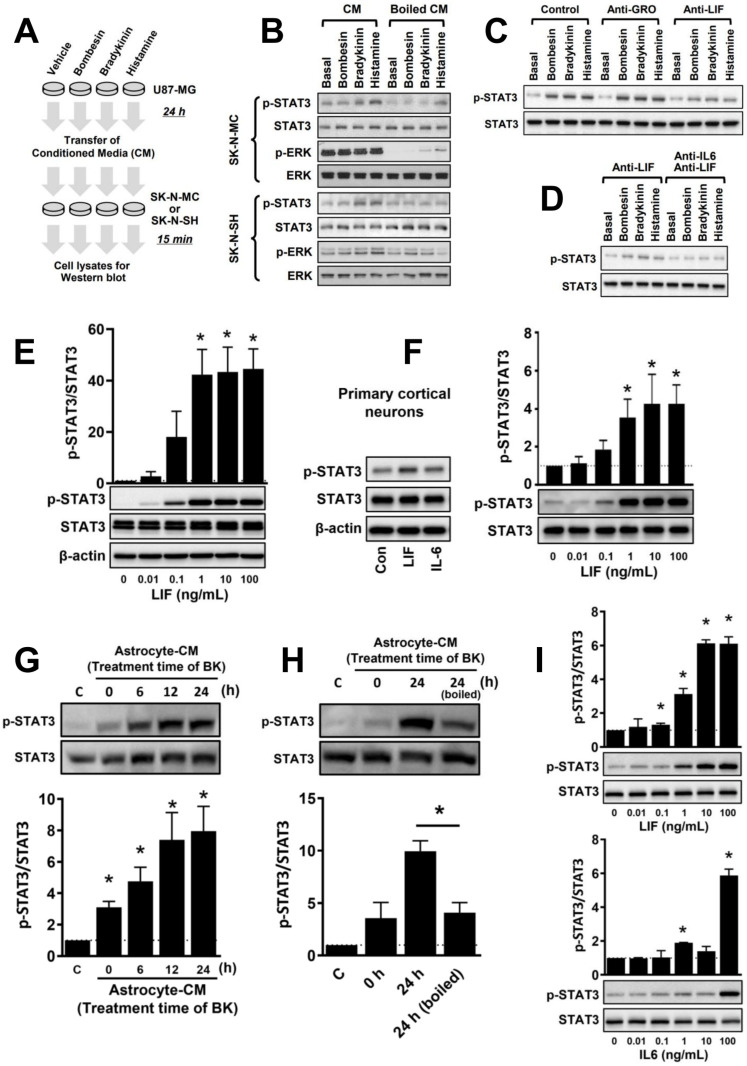
Activations of G_q_-coupled receptors lead to the production of soluble factor(s) that stimulate STAT3 phosphorylation in different cell types. (**A**) Schematic illustration of the preparation and testing of conditioned media from U87-MG cells. (**B**) Conditioned media (with or without being subject to heat treatment at 96 °C) from U87-MG cells were used to stimulate SK-N-MC and SK-N-SH cells for 15 min, followed by SDS-PAGE and the immuno-detection of phosphor-STAT3 and phospho-ERK. (**C**) Conditioned media from U87-MG cells were incubated with or without anti-GRO (negative control) or anti-LIF antisera and assayed on SK-N-MC cells as in panel B. (**D**) SK-N-MC cells were treated, as in panel C, in the presence of both anti-LIF and anti-IL-6 antisera. (**E**) SK-N-MC cells were stimulated with increasing concentrations of LIF and then assayed for STAT3 phosphorylation. (**F**) Primary mouse cortical neurons were stimulated with various concentrations of LIF and then assayed for STAT3 phosphorylation. (**G**) Primary mouse astrocytes were incubated with bradykinin of increasing duration (0, 6, 12, 24 h without serum), the resulting conditioned media were collected and then applied to primary mouse cortical neurons for 15 min, followed by immuno-detection of phosphorylated STAT3 in the recipient neuronal cells. (**H**) The conditioned medium collected from primary mouse astrocytes (having 24 h bradykinin treatment) induced STAT3 phosphorylation effectively, while heat treatment at 96 °C significantly diminished its STAT3-activating function. (**I**) Primary mouse astrocytes were stimulated with various concentrations of LIF and IL-6 and then assayed for STAT3 phosphorylation. * Application of LIF, IL-6, and conditioned media from bradykinin-treated primary astrocytes led to significant stimulatory phosphorylation of STAT3 in the recipient cells (* *p* < 0.05; *n* = 3).

**Figure 5 ijms-25-13079-f005:**
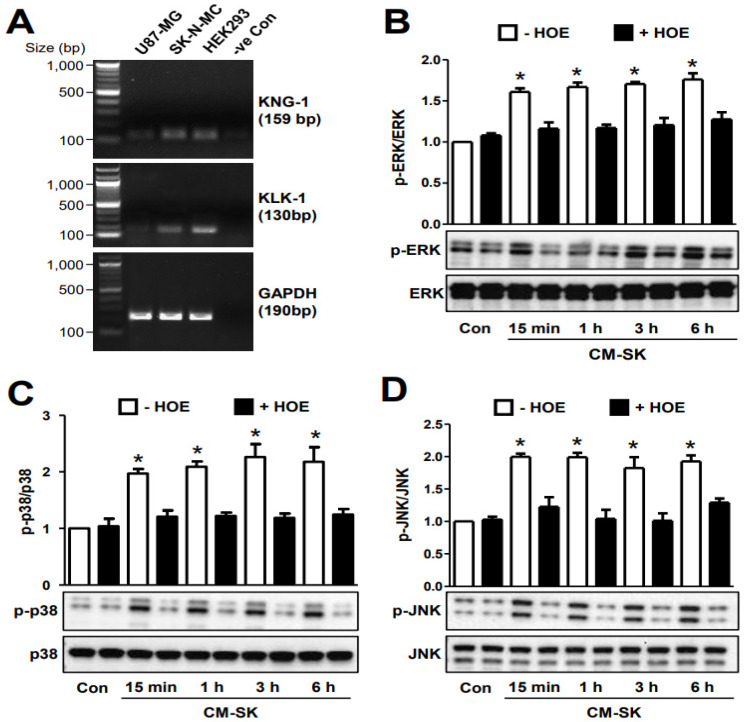
The KNG-1/KLK-1 mRNA transcripts were readily detected in neuronal SK-N-MC cells, and the corresponding conditioned media could act through the bradykinin B_2_ receptor to stimulate various MAPK subtypes in U87-MG cells. (**A**) Detections for mRNAs of KNG-1 and KLK-1 from the total RNA extracts of U87-MG cells, SK-N-MC cells, and HEK293 cells (+ve control) by reverse transcription PCR, with the GAPDH signal served as sample loading controls. (**B**) Conditioned media obtained from SK-N-MC cultures (CM-SK) under serum removal of 15 min, 1 h, 3 h, and 6 h were applied to U87-MG cells with or without HOE-140 pretreatment, and the resulting U87-MG cell lysates were subjected to SDS-PAGE and immunodetection for stimulatory phosphorylated ERK (p-ERK) and total ERK (ERK). Immunodetection of (**C**) phosphorylated/total p-38 and (**D**) phosphorylated/total JNK were performed in a similar manner as that shown in panel (**B**). * Conditioned media obtained from SK-N-MC cultures of different indicated collection time-points were all capable of inducing stimulatory phosphorylation of ERK, p38, and JNK significantly, as compared with the control group (* *p* < 0.05; *n* = 3), with HOE-140 (B_2_R antagonist) pretreatment suppressing the activation of all three of these kinases.

**Figure 6 ijms-25-13079-f006:**
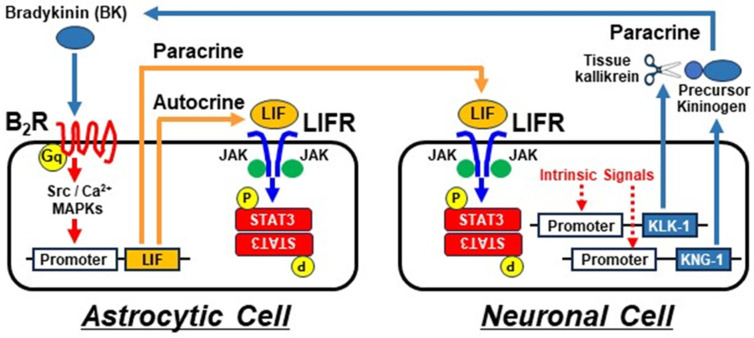
A schematic diagram of the possible autocrine–paracrine communication between astrocytic cells and neuronal cells. Constitutively expressed components of the kinin-generating system in neuronal cells contribute to the release of bradykinin, which can act on bradykinin B_2_ receptors (B_2_R) present in astrocytic cells. The resulting B_2_R/G_q_ activation triggers a sequential signaling event (including the upstream Src and Ca^2+^ signals and the downstream MAPK kinase cascades) to induce the production of secretory LIF proteins. LIF will subsequently act on its receptors (LIFR), expressed in astrocytic cells themselves (autocrine pathway) or on LIFR that are expressed in neighboring neuronal cells (paracrine pathway). In both cell types, LIFR will recruit the tyrosine kinase, JAK, which eventually triggers the stimulatory phosphorylation of STAT3. The final fates of developmental changes in astrocytic cells and neuronal cells may rely on their epigenetic characteristics, as well as additional signaling inputs due to differential receptor expression profiles.

## Data Availability

The data presented in this study are available on reasonable request from the corresponding author.

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
