# Peer review of "Activation of Bradykinin B2 Receptors in Astrocytes Stimulates the Release of Leukemia Inhibitory Factor for Autocrine and Paracrine Signaling"

_ijms, 2024, doi:10.3390/ijms252313079_

Round 1

Reviewer 1 Report

Comments and Suggestions for Authors

In the central nervous system, interactions between neurons and glial cells regulate neurotransmission, energy metabolism, extracellular ion balance, and neuroprotection. This study demonstrates that bradykinin, a proinflammatory neuropeptide, is detected by astrocytes, leading to cytokine secretion that affects neurons. Bradykinin and other Gq-coupled receptor ligands triggered Ca2+ mobilization in astrocytes, resulting in the release of leukemia inhibitory factor (LIF) and interleukin-6 (IL-6). The B2 receptor antagonist, HOE-140, blocked bradykinin-induced Ca2+ mobilization and MAPK activation in astrocytes. Conditioned media from bradykinin-treated astrocytes activated STAT3 in neurons. Neurons were found to produce bradykinin, as their conditioned media stimulated MAPKs in astrocytes in a HOE-140-sensitive manner. These findings highlight paracrine signaling between neurons and astrocytes, involving Gq-coupled receptor ligands and cytokines like LIF.

Some minor problems:

1.“μl” should be “μL”: Please ensure consistent and correct use of units throughout the manuscript. In lines 137-139, 141, 143, 152, 154, and possibly elsewhere, "μl" should be capitalized to "μL" as per standard scientific notation. Correcting this enhances clarity and adherence to scientific conventions.

2. Line 168, GAPDH primer sequences should be included in RT-PCR section: In the methods section, specifically line 168, the primer sequences for GAPDH, a widely used reference gene, should be provided. Including this information is important for reproducibility and allows readers to validate the experiment or compare with their own results.

3. Formatting of statistical annotations: In several instances (lines 214, 219, 253, 344, 354, etc.), the p-value notation "(p<0.05)" should be revised to "(* p < 0.05)" to maintain consistency with the rest of the manuscript and standardize the way statistical significance is presented across the text. This will improve readability and visual uniformity.

4. Avoid including supplementary figures (S1, S2) in the main text: The supplementary figures S1 and S2 are currently referenced or shown in the main text, which should be avoided. Supplementary data are intended to provide additional information without interrupting the flow of the primary content. Authors should either fully integrate relevant data into the main figures or ensure that supplementary figures are only referred to in the supplementary section.

5. Figure 5 panels should appear on a single page: For better presentation and reader experience, all panels of Figure 5 should be consolidated on a single page. This will enhance visual flow and allow for easier interpretation of the data. Splitting figure panels across multiple pages disrupts comprehension and can confuse readers.

Author Response

Please see the attachment, thank you !

Reviewer 2 Report

Comments and Suggestions for Authors

The study investigates signalling cascade between neurones and astrocytes which operates using bradykinin and LIF. The authors report that bradykinin, acting via Gq coupled B2 receptors stimulates release of LIF from astrocytes which seems to act on receptors on both neurones and astrocytes, while bradykinin seems to be of neuronal origin. The study uses mainly cell lines, Western blot analysis and Ca2+ imaging.

There are many issues which this paper, both conceptual and technical.

Conceptually, it appears that paper is largely based on experiments on cell lines, largely of glioblastoma/astrocytoma origin or unclear origin for "neuronal" cell line. I simply disagree that such inferences can be allowed without a direct proof in real cells. But if all this can be shown in real astrocytes and neurones, why do we care about tumour cells?

2.1. Materials (101) “HEK293, U87-MG, 1321N1, SK-N-SH, and SK-N-MC cells were …” – none of those are actually neither astrocytes nor neurones. U87 and 1231N1 are cells of glioblastoma/astrocytoma origin. The biological identity of SK cells is obscure. HEK are even less relevant. Possibly for some simple and very basic studies where protein-protein interactions are studied or something of that kind, such cell lines are suitable. But when it comes to signalling between differentiated neurones and astrocytes in the brain these cell lines are of hardly any  use. They have a completely different gene expression profiles, they do not possess appropriate physiology or morphology and are nothing else but tumour cells with some traits resembling their origins. Therefore the only part of the paper which really deals with neurones and astrocytes is where primary cultures are used. Even those are from embryonic brains and we know that neither neurones nor astrocytes are mature in such preparations.

In my opinion, the whole paper needs to be re-written to scale down claims that the described signalling occurs in neurones and astrocytes, unless the authors undertake another set of experiments in native cells and confirm all their results there. Which then will make most of the current manuscript obsolete because we don’t really care about tumour derived cell lines.

One obscure aspect of this study is the mechanism by which ILF is secreted. Figure 3C shows that the peak of LIF concentration occurs at ~24 hours, so clearly this is not just release but induction at genomic level. So the link to Ca2+ is then not direct, as one could suggest from the first part where Ca2+ increases are studied. When bradykinin was studied, MAPKs were activated within 10 min and PCK inhibition was not particularly effective. Overall it is not clear how bradykinin works and why the time lag is so great.

Specific comments are below.

1.      Line 38. Astrocytes are not immune cells. Re-write.

2.      Results. Section 3.1 has to be changed to reflect what was really happening. Section starts with experiments on HEL cells. To me the relevance of this section is unclear, because in HEK cells and astrocytes situation can be completely different, completely different genes are expressed, they are localised differently etc. Title has to change. HEK are not astrocytic cells and neither are GBM cell lines. In fact even though data from HEK cells are interesting they are not directly relevant to astrocytes. I would put them into a supplement if use at all. In any case we are mainly interested in cells other than astrocytes. Cell lines are from glioblastoma/astrocytoma and they are not astrocytes either, this has to be acknowledged, calling them perhaps “models of astrocytes”. The most important is data from astoryctes and then it can be compared with cell lines. Data from HEKs could perhaps go into supplement.

3.      Figures/bar charts. I could not see the n numbers in all cases, whether they were technical or biological replicates. Change graphs to dot/bar charts and write clearly the n numbers.

4.      Remove all instances where U87 and 1321 are called “astrocytes” i.e. pg 6 line 220. In Fig 2b line 1321N1 does not respond to ATP which is one of the most reliable signalling molecules to trigger Ca2+ increases in astrocytes.

5.      Concentrations of drugs are not shown on graphs nor are n numbers. Whe there are replicates  it has to be stated, what was used for statistics (and the test).

6.      Astrocytic cultures are likely to contain microglia, it would be useful to show IBA1 staining.

7.      What is plotted on figure S2A, what are the units?  Are there error bars? They seem to be so small, that it is hard to believe that, given how unreliable western blots usually are. What are the n numbers?

So finally, conclusion that “Hence, neuron-derived bradykinin could be one of the major sources to initiate the astrocyte-derived LIF production.” Cannot be made based on the use of these odd cell lines. If the authors can show that native neurones (at least in slices) release bradykinin which drives LIF secretion from astrocytes, the whole study comes into question.

Author Response

Please see the attachment, thank you !

Reviewer 3 Report

Comments and Suggestions for Authors

In the manuscript from Lu et al., the authors describe investigations into the autocrine/paracrine signaling between glial cells and neurons in vitro related to inflammatory signaling via bradykinin, LIF, STAT3, etc.  While the results are interesting, the manuscript is seriously lacking information in order to appropriately evaluate the study design and results and cannot be published in its current form.  My major concern is that the manuscript is seriously lacking in detailed methods for the study with several methods not described at all in the manuscript.  Supplementary figures are provided, but not supplementary materials describing any of the methods for these supplemental experiments are described.  Without appropriate description of the methods, I cannot fully evaluate the merit and validity of the study.  Additional detailed comments are outlined below:

1)  1)      While the introduction states what is reported in the study, it would be beneficial to tie in all of the background information provided in the introduction into a statement of the rationale/hypothesis that was investigated for the study.

1)      While the introduction states what is reported in the study, it would be beneficial to tie in all of the background information provided in the introduction into a statement of the rationale/hypothesis that was investigated for the study.

2) Please provide references for the primary cell line isolation procedures, if available.

3) The manuscript is disorganized with some methods briefly described in the results without detailed description in the methods.  The description of methods in the results makes it difficult to absorb the data being presented in the results section. While the general culturing of cells are described, there is no description of transfection, treatment with specific ligands, treatment with anti-serum, boiling, etc that is alluded to throughout the results.  Timing of experiments and treatments are not described, making it impossible to evaluate the study design and results.  No description of stat3 antibodies, Anti-LIF antiserum, anti-GRO antiserum, anti-IL-6 antisera, etc.

14)      Please clarify kits used for cytokine multiplex assay.  The methods indicates kits from Merck Millipore, but references indicate Affymetrix and Merck Millipore.  For transparent methods, please indicate the type of kits and the cytokines measured in each, particularly since both referenced manuscripts are not open access publications.

25)      Please identify manufacturer of FLIPR instrument

36)      Was software used for primer design?  If so, please describe.  Also, include primer sequences or source for GAPDH primers.

47)      Ref(s) for WB procedure?  Please clarify proteins measured by WB in the methods.  Also, clarify system used to measure chemiluminescence and how band density was measured.

68)      No materials and methods are provided for the Gα subunit transient transfection (I presume) or the 23 GPCR ligand stimulation described in Section 3.1 of the results.

79)      RRIDs should be provided for all antibodies, cell lines, software for proper identification of materials.

810)      What is the rationale for using HEK293 cells when looking at neuroinflammation?

111)      Considering interest in autocrine function, why was conditioned astrocyte media not applied to naïve cells to determine autocrine LIF responses?

212)      The Discussion lacks any description of limitations of the study.  Differences between primary and immortalized/cancer-derived cell lines is not discussed and is a crucial point to make.  How could the cancer state of the human derived cell lines impact the processes that are being evaluated in the study? While cell culture is indeed a useful tool to evaluate the mechanisms described in the study, the authors mention the importance of cross-talk between cell types – which is lacking in independent cell cultures of glia or neuronal-like cells. Could these studies be done in co-culture, ex vivo tissue slices, in vivo experiments to verify that this signaling actually occurs in intact tissue?  How could these finding be applied to study in animal models of the diseases mentioned – ie PD, AD, ALS?

Author Response

Please see the attachment, thank you !

Round 2

Reviewer 3 Report

Comments and Suggestions for Authors

1) Overall, the manuscript still lacks a clear description of the study design due to a lack of clarity/information in the methods.  All detailed methodology should be described in the Materials and Methods section of the manuscript.  While the authors have added several details regarding the methods over the previous submission, much of this has been added to the Results, Figure Legends, or Supplemental Figure Legends (and some in the Discussion) instead of the Methods section.  Spreading out the information in this way makes it difficult for readers to find the specific details that would be needed to understand or potentially replicate the assay.  Please carefully review the manuscript and ensure that methods are provided for ALL experiments described.  I have done my best to try to capture what appears to be missing, but the manuscript is very dense and I may have missed some details. I would suggest adding specific sections or subsections to the Methods to describe specific details regarding the following:

-Transfection:  all Gα subunits transfected should be listed in the Methods, brief details on how much plasmid was transfected (dependent on plate well-size per protocol), % confluence when transfected, how long after transfection was the ELISA performed, MTR receptor source and transfection conditions, details on melatonin source and conditions for stimulation (concentration and time applied).

-GPCR stimulation:  add a table with all ligands and concentrations used, provide details on % confluence when stimulated, amount of time ligands were applied prior to detecting calcium levels. 

-Bradykinin receptor inhibition:  provide details regarding % confluence when exposed, concentration(s) and amount of time HOE-140 was applied prior to stimulation with bradykinin, concentration and amount of time bradykinin was applied prior to measuring calcium levels

-MAPK pathway stimulation/inhibition:  provide details regarding % confluence when bradykinin applied, concentration(s) and amount of time bradykinin was applied prior to measuring ERK, p38, JNK and phosphorylated species, details on HOE-140 exposure conditions (concentration, time, etc), details on specific protein kinase inhibitors name/target/etc, concentration(s)/time exposed prior to bradykinin and bradykinin exposure details.

-Conditioned media:  provided details regarding concentration(s) of agonists, time of exposure, culture conditions, etc for the production of astrocyte conditions media (from U87-MG cells) and culture conditions/application of conditioned media to SK-N-SH and SK-N-MC cells.  Was assay performed immediately after 15 min incubation or at some later time point?  Please clarify in methods.  How was conditioned media boiled to inactivate soluble factor(s)?  Similar details need to be provided for similar experiments completed with primary astrocytes/neurons, SK-N-MC conditioned media/HOE-140 inhibition/U87-MG cell exposure.

-Anti-sera neutralization: provide details on amount/concentration of anti-sera added to the conditioned media and for how long before applying antiserum-treated conditioned media to SK-N-MC cells.

-LIF stimulation:  culture conditions/concentration/amount of time of exposure to LIF, source of LIF?

-Immunofluorescence – no details are provided for antibodies (NeuN, GFAP), cell prep, immunofluorescence staining protocol, imaging, etc.

Additional information to be added to the methods:  a) List all cytokines measured in the Merck Millipore multiplex; b)  details must provided for GFAP antibody for Western blot

As the transfection experiments and immunofluorescence are exclusively presented in Supplementary materials, the authors could consider adding a Supplementary Methods section in the Supplementary Materials to provide the specific details for these studies.   

2) The limitations added to the beginning of the Discussion should be moved to the end of this section and combined with a conclusions paragraph.  While at least two summary statements are made in the Discussion (“Taken together, …” at Lines 581 and 627), some “overall” conclusions for the study and suggested future directions would improve this section of the manuscript.

Round 3

Reviewer 3 Report

Comments and Suggestions for Authors

I wish to thank the authors for their careful consideration of the previous comments and revision of the manuscript.  Overall, the presentation of the manuscript is much improved and the transparency of the methods is much better. 

One minor comment that should be addressed before publication is that the antibody concentrations/dilutions for immunofluorescence (NeuN, GFAP, MAP2, and Alex Fluor secondary antibodies; Supplementary Methods) and Western blot (GFAP, B-tubulin, LIFR - Supplementary Methods; B1R, B2R, ERK, p-ERK, p38, p-p38, JNK, p-JNK, STAT3, p-STAT3 - Main Methods) are not provided.
